# Prevalence of Common Alleles of Some Stress Resilience Genes among Adolescents Born in Different Periods Relative to the Socioeconomic Crisis of the 1990s in Russia

Svetlana V. Mikhailova [1],[*], Dinara E. Ivanoshchuk [1], Evgeniy A. Yushkevich [1], Ahmad Bairqdar [1],
Maksim S. Anisimenko [1], Liliya V. Shcherbakova [2], Diana V. Denisova [2] and Pavel S. Orlov [1]

[1] Institute of Cytology and Genetics, Siberian Branch of Russian Academy of Sciences (ICG SB RAS),
10 Prospekt Ak. Lavrentyeva, Novosibirsk 630090, Russia
[2] Institute of Internal and Preventive Medicine—Branch of ICG SB RAS, 175/1 Borisa Bogatkova Str.,
Novosibirsk 630089, Russia
[*] Correspondence: mikhail@bionet.nsc.ru

**Abstract:** Social stress is common among people and is considered one of the causes of the declining birth rate. Predisposition to stress and stress-induced disorders is largely determined genetically. We hypothesized that due to differences in stress resistance, carriers of different genetic variants of genes associated with stress resilience and stress-induced diseases may have dissimilar numbers of offspring under conditions of long-term social stress. To test this hypothesis, a comparative analysis of frequencies of seven common polymorphic regions [exon 3 variable number of tandem repeats (VNTR) of the *DRD4* gene, rs4680 of *COMT*, STin2 VNTR and the 5-HTTLPR (rs774676466) insertion/deletion polymorphism of *SLC6A4*, rs4570625 of *TPH2*, rs6265 of *BDNF*, and rs258747 of *NR3C1*] was performed on standardized groups of randomly selected adolescents born before, during, and after severe socioeconomic deprivation (the crisis of the 1990s in Russia). There were significant differences in frequencies of "long" alleles of the *DRD4* gene ($p = 0.020$, $\chi^2 = 5.492$) and rs4680 ($p = 0.022$, $\chi^2 = 5.289$) in the "crisis" group as compared to the combined "noncrisis" population. It is possible that the dopaminergic system had an impact on the successful adaptation of a person to social stress.

**Keywords:** social stress; stress resilience; polymorphism; *DRD4*; *COMT*; *SLC6A4*; *TPH2*; *BDNF*; *NR3C1*

## 1. Introduction

Social stress (the sociological term), i.e., the stress of social interactions (the biological term) is the most common type of stress in humans; stress-induced mental disorders are common in the general population [1]. When a stressor cannot be eliminated, the stress becomes chronic. Under this condition, the hypothalamic–pituitary–adrenal (HPA) axis is constantly activated, which is accompanied by an elevated production of cortisol, the main "stress hormone" in humans. Prolonged hypercortisolemia and consequent chronic neuroinflammation in the brain reduce the sensitivity of glucocorticoid receptors (GRs) in the pituitary and hypothalamus. In turn, increasing steroid resistance impairs the feedback mechanism within the HPA axis [2,3]. Chronic stress can cause anxiety and mood disorders and provoke attention deficit/hyperactivity disorder (ADHD) [4] and depression [5]. It is believed that chronic stress affects the human reproductive system through the interaction of the HPA axis with the hypothalamic–pituitary–gonadal axis. The stress-activated cortisol release desensitizes the pituitary to gonadoliberin, thus reducing the secretion of gonadotropins (follicle-stimulating hormone and luteinizing hormone) [6,7]; these changes can affect sexual activity [8]. Under prolonged stress, the interaction of the reproductive and stress axes worsens both the reproductive health of a person and his/her

emotional well-being [9]. This problem can lead to subdominant reproductive behavior (a decline in the number of children or a decision not to have them). In Russia, during 1990–1998, there were sharp changes in the social, economic, political, and cultural spheres and in the scale of values; these alterations gave rise to contradictions within public consciousness and a decline in social well-being, and consequently an increase in social stress. During the 1991–2001 period, there was the largest drop in the birth rate in the urban population of Russia in the second half of the 20th century [10]. The number of first-time diagnoses of nonpsychotic disorders (e.g., acute reactions to stress, adaptive neurotic reactions, and neuroses), as determined under outpatient observation, fell threefold from 1995 to 2016 according to the statistics of the Russian Ministry of Health [11] (p. 31). Heritability of many stress-induced disorders and stress-related personality traits has been proven: for risk tolerance, it ranges from 20% to 60% [12]; for anxiety disorders, it is estimated in the range of 30–50% [13]; and for ADHD, this parameter reaches 75% [14]. Risk of anxiety disorders positively genetically correlates with risks of major depressive disorders, schizophrenia, ADHD, and neuroticism affecting subjective well-being [15]. Determining the genetic causes of stress-induced disorders is sometimes difficult because some genetic variants are associated with disorders only during stressful events [16]. Among already identified anxiety-related genes, there are genes encoding enzymes, hormones, neurotransmitters, their transporters, and receptors [17–19].

Serotonin (5-HT) is synthesized in serotonergic neurons. The serotonergic system is inhibitory for the nervous system, mediates stress responses, and helps to maintain the feeling of calm. Increased synaptic 5-HT availability in the central nervous system is linked with reduced anxiety [20]. The *SLC6A4* gene (solute carrier family 6 member 4) encodes serotonin transporter 5-HTT, which is responsible for the reuptake of 5-HT from the intrasynaptic space and regulates the biological actions of 5-HT. *SLC6A4* is the most widely studied gene in the field of the genetics of stress [21,22]. It is reported to be associated with ADHD [18,23]. Common alleles of an insertion/deletion (indel) polymorphism in the promotor of *SLC6A4* (5-HTTLPR, serotonin transporter-linked polymorphic region, rs774676466)—long (L) of 16 repeated elements and short (S) of 14 repeated elements of 43 bp—affect the promoter activity. The S variant is associated with lower transcriptional efficiency in *SLC6A4*, resulting in decreased serotonin uptake compared to L [24]. The S allele correlates with anxiety disorders and some anxiety-related traits according to several studies and meta-analyses [25–27]. It has been found that individuals carrying lower-expression variants of rs774676466 are more susceptible to various types of stress [28]. Nonetheless, there are still debates regarding the clinical significance of this promoter variant [29]. Variable numbers of tandem repeats (VNTRs) in intron 2 (STin2) of *SLC6A4* denote a variable number (9, 10, or 12) of 16–17 bp tandemly repeated units. This VNTR site functions as a transcription-regulatory domain, and transcription factors CTCF and YB-1 can bind to this region and influence *SLC6A4* expression [30]. The allele featuring 12 repeats has the highest transcriptional activity [31]. This variant is reported to be associated with mood disorders [32].

The *TPH2* gene encodes an enzyme called tryptophan hydroxylase 2, which catalyzes the rate-limiting step of serotonin biosynthesis in the brain [33]. In combination with uncontrollable stress, *TPH2* variants predispose to anxiety-, aggression-, and depression-related personality traits [34]. Single-nucleotide polymorphism (SNP) rs4570625 ($-703$G/T) is located in the upstream regulatory region of the *TPH2* gene. Carriers of the T allele of rs4570625 have been found to have smaller amygdala and hippocampal volumes and a higher reward dependence than do homozygotes for the G allele [35]. The minor T allele of rs4570625 is associated with low anxiety, low aggressiveness, and lower prevalence of anxiety disorders [36].

The dopamine system is responsible for voluntary movement, motivation, reward, cognitive function, remembering negative experiences, and reproductive behavior [37]. Dopamine is produced by certain peripheral organs and in the central nervous system where it acts as a neurotransmitter. The pathophysiology of Huntington's disease, Parkin-

son's disease, schizophrenia, ADHD, and addictions is associated with disturbances of dopamine metabolism [37]. Dopamine receptors are important mediators of the HPA axis [38]. The *DRD4* gene encodes the D4 dopamine receptor. An SNP in its promoter correlates with low stress resilience [22]. VNTR in exon 3 of *DRD4* is located in a region that encodes the third intracytoplasmic loop of the receptor. There is a 48 bp sequence representing a 2- to 11-fold repeat; of the 16 encoded amino acid residues, 5–6 are prolines. VNTR alleles with less than seven repeats are traditionally referred to as "short", and those with seven or more repeats as "long". The most common globally is the 4R allele (frequency ranging from 0.16 to 0.96). The 7R allele occurs at a frequency of ~0.48 in the Americas but only 0.02 among Asian populations, and other variants are rare [39]. The 7R variant has a lower affinity for dopamine and blunts dopamine signaling [40]. The 7R allele is reported to correlate with high novelty seeking independently of ethnicity, sex, and age [41]. Carriers of alleles 2R and 7R have stronger reward processing compared to noncarriers [42]. Carriers of the 7R allele exhibit lower cortisol stress responses after the Trier social stress test [43], but they seem to have stronger brain activity in response to negative emotional stimuli as compared to individuals with the 4R allele [44]. It has been supposed that the 7R variant has reached high frequency in human populations via positive selection [45,46]; however, this opinion is disputed [47]. Long alleles of this VNTR have been shown to predispose to ADHD [48]. It has been suggested that *DRD4* may not be an attention deficit risk gene but may serve as a plasticity gene; accordingly, individuals with the long alleles show heightened selective attention to high-priority items [49,50].

The *COMT* gene codes for catecholamine-O-methyltransferase, an enzyme taking part in the degradation of dopamine. It catalyzes O-methylation of 3,4-dihydroxyphenylacetic acid, which is an intermediate of dopamine [51]. The membrane-bound form of COMT is mainly expressed in the prefrontal cortex, a region involved in fear inhibition. It is COMT that is responsible for dopamine clearance in this brain region because of low dopamine transporter expression there [52]. The *COMT* gene has been implicated in panic disorders [53]. Rs4680 (G/A, Val158Met) is one of the most investigated SNPs in human neuroscience. The Met-allele lowers COMT activity by approximately 40%, and therefore Met-allele carriers have higher brain dopamine levels. This amino acid substitution influences the thermal stability and activity of COMT, such that the COMT activity in the human prefrontal cortex is approximately 35–50% lower in Met homozygotes than in Val homozygotes. Heterozygous carriers have an intermediate phenotype. As a result, dopamine signaling is likely enhanced in Met carriers compared to Val carriers [54,55]. It is thought that under stress, people with G (Val) alleles may have better dopaminergic transmission and show better performance than people with Met alleles; accordingly, the Met158 allele is associated with anxiety [56]. Nonetheless, the literature is ambiguous as to which allele of rs4680 poses a risk of clinical anxiety and other stress-induced disorders; some studies suggest that it is the G allele or the G/G genotype in the general population [57,58] or in males only [59], while other articles indicate that the risk allele is the Met variant but only among females [60]. Some researchers have failed to find an association of this SNP with a risk of anxiety disorders [61]. Ethnicity affects the correlation of the Val allele with psychiatric disorders; it is associated with ADHD and panic disorder among whites but not among Asians [62].

The *BDNF* (brain-derived neurotrophic factor) gene is related to the neurodevelopment system. BDNF plays a crucial role in synaptogenesis and in the consolidation and maturation of synapses; it can regulate long-term potentiation, which is a sustained increase in excitatory synaptic efficiency that underlies learning and memory [63]. A haploinsufficiency of the *BDNF* gene has been shown to cause profound dysfunctions of adaptive behavior in humans [64]. A correlation of *BDNF* with risky behavior was found in a genome-wide association study [12]. Plasma levels of BDNF are significantly lower in patients with panic disorder than in a control group [65]. B-cell lines homozygous for the L allele of 5-HTTLPR in *SLC6A4* show a dose-dependent decrease in 5-HT uptake after exposure to BDNF [66]. SNP rs6265 G/A in exon 13 of the *BDNF* gene (Val66Met) is widely studied as

a susceptibility factor for stress- and anxiety-related disorders. The BDNF protein resulting from the Met allele undergoes inefficient secretion and intracellular trafficking [67]. The *BDNF* A/A genotype may diminish the plasma BDNF level and increase trait anxiety in panic disorder [65]. A hypothesis has been advanced that the impairment of endogenous BDNF activity by the 66Met variant potentiates sensitivity to stress and as a result to stress-induced diseases. Thus, BDNF may promote the encoding of fear and trauma by facilitating neuronal plasticity [68]. The A allele has been shown to be more common in stress-induced disorders [22,69–73]. There are also some contradictory data [74].

The *NR3C1* gene (nuclear receptor subfamily 3, group C, class 1) encodes the gluco-corticoid hormone receptor that binds cortisol. GRs are present throughout the brain, and the balance in the functioning of the mineralocorticoid receptor and GR in hippocampal neurons is critical for stress responsiveness and behavioral adaptation. This receptor is located in the cytoplasm as a component of a multiprotein complex. After a stress-induced release of cortisol, the GR binds to the latter and moves to the nucleus, where it can function as a transcription factor or as a regulator of other transcription factors [75,76]. A correlation has been demonstrated between the presence of various polymorphisms of the *NR3C1* gene and the response of the HPA axis to psychosocial stress [77,78]. Predeployment GR pathway components are factors predisposing to severe symptoms of post-traumatic stress disorder [79]. SNP rs258747A/G is located in the 3′ untranslated region of the *NR3C1* gene. This SNP is associated with lower cortisol levels and post-traumatic symptoms both separately and in terms of gene–environment interaction [80,81].

In the present work, a comparative analysis of genotypes of adolescents born in the periods before, during, and after the crisis of the 1990s (years of DNA sample collection: 1999, 2009, and 2019, at ages 14–17 years) in Novosibirsk city (Russia) was performed. The STin2 VNTR and 5-HTTLPR (rs774676466) indel polymorphism of the *SLC6A4* gene (serotonin transporter), VNTR in exon 3 of the *DRD4* gene (dopamine receptor), as well as SNPs rs4680 in the *COMT* gene (dopamine catabolism enzyme), rs4570625 in the *TPH2* gene (serotonin biosynthesis enzyme), rs6265 in the *BDNF* gene (neurotrophic growth factor), and rs258747 in the *NR3C1* gene (cortisol receptor) were analyzed. The obtained data allowed us to assess genetic differences between the generations born during the crisis and consequently to identify the genes that may affect reproductive behavior in the generation of their parents.

## 2. Materials and Methods

The study protocol was approved by the ethical committee of the Institute of Internal and Preventive Medicine, a branch of the Institute of Cytology and Genetics of the Siberian Branch of the Russian Academy of Sciences, Novosibirsk, Russia (protocol No. 7, approved on 22 June 2008). Blood samples were obtained by the Institute of Internal and Preventive Medicine during a standardized medical cross-sectional examination (monitoring of mental, physical, and emotional state), which has been carried out since 1989 every 5 years. Random representative groups of unrelated schoolchildren aged 14–17 years of both sexes (male:female ratio of 40:60) were formed in randomly selected grades of 10 out of 20 secondary schools in the Oktyabrsky district of Novosibirsk city (Russia), which is a typical area in the industrial center of Western Siberia in terms of the number of industrial enterprises and the number of cultural, medical, and educational institutions, and the ethnic composition of the inhabitants. In the selected school grades, a complete survey of schoolchildren was conducted, and the response rate was 95%. Each time, of the approximately 7000 adolescents living in the study area, blood specimens were collected from approximately 10%, which ensured a representation of the populational sample. These adolescent groups have been described previously [82]. In this work, the sampling was performed in the years 1999, 2009, and 2019. Accordingly, three independent groups of adolescents were compiled: those born in 1982–1985 (group 1, n = 451); those born in 1992–1995 (the period of the socio-economic crisis of the 1990s in Russia: group 2, n = 694); and subjects born in 2002–2005 (group 3, n = 483). Each group consisted of at least 95%

whites; the ethnicity of the individuals was determined using a questionnaire and an additional cross-sectional survey to identify the nationality of the ancestors.

Blood samples were stored at −20 °C until genomic DNA was isolated from blood leukocytes by standard phenol–chloroform extraction [83]. For PCR analysis of DNA fragments of genes *DRD4*, *SLC6A4*, and *COMT*, direct and reverse primers (Biosset, Novosibirsk, Russia) were employed (Table 1). For the genotyping of the rs4680 polymorphism (*COMT* gene), the PCR product was digested by means of restriction endonuclease BstHH I (SibEnzyme, Novosibirsk, Russia). Sizes of the PCR products and restriction fragments were estimated by electrophoresis in a 5% polyacrylamide gel.

**Table 1.** Conditions for PCR and restriction analysis for the VNTR of exon 3 in *DRD4*, VNTR STin2 and rs774676466 in *SLC6A4*, and rs4680 of *COMT*. R: the number of repeats.

| Gene, Polymorphism | PCR Primers | Annealing Temperature, °C | Amplicons' Lengths, bp | Restriction Fragment Lengths, bp |
|---|---|---|---|---|
| *DRD4*, VNTR exon 3 | 5′-AGGTGGCACGTCGCGCCAAGCTGCA-3′<br>5′-TCTGCGGTGGAGTCTGGGGTGGGAG-3′ | 66 | 462 (8R), 414 (7R), 366 (6R), 318 (5R), 270 (4R), 222 (3R), 174 (2R) | – |
| *SLC6A4*, rs774676466 | 5′-GGCGTTGCCGCTCTGAATGCC-3′<br>5′-CAGGGGAGATCCTGGGAGAGGT-3′ | 62 | 270 (L), 182 (S) | – |
| *SLC6A4*, VNTR STin2 | 5′-GTCAGTATCACAGGCTGCGAG-3′<br>5′-TGTTCCTAGTCTTACGCCAGTG-3′ | 60 | 299 (12R), 267 (10R), 250 (9R) | – |
| *COMT* rs4680 | 5′-GGGCCTACTGTGGCTACTCAGCTGT-3′<br>5′-GGCATGCACACCTTGTCCTTCG-3′ | 64 | 148 | BstHH I,<br>A/A: 148<br>A/G: 148, 126, 22<br>G/G: 126, 22 |

Genotyping of rs6265, rs258747, and rs4570625 was conducted by real-time PCR on a LightCycler 96 instrument (Roche, Basel, Switzerland). Flanking oligonucleotides and TaqMan probes were selected in Vector NTI Advance 11.0 software (Thermo Fisher Scientific, MA, USA). The flanking oligonucleotides and complementary probes labeled with dyes FAM and HEX and with the BHQ1 fluorescence quencher are listed in Table 2.

**Table 2.** Oligonucleotide and TaqMan probe sequences used for real-time PCR analysis of rs6265, rs258747, and rs4570625.

| SNP | Oligonucleotides | Probes |
|---|---|---|
| *BDNF* rs6265 | 5′-CCAAGGCAGGTTCAGAGGCT-3′<br>5′-TTCATGGGCCGAACTTTCTGG-3′ | [FAM] TCATCCAACAGCTCTTTATCACGTGTT [BHQ1]<br>[HEX] TCATCCAACAGCTCTTTATCATGTGTT [BHQ1] |
| *NR3C1* rs258747 | 5′-ATCATCATGTGCACCAAGTAT-3′<br>5′-ATACTCTGATTGAGGGTACAA-3′ | [FAM] ACATAGTATTTTTCTTATTCACATTGT [BHQ1]<br>[HEX] ACATAGTATTTTTCTTATTCACGTTGT [BHQ1] |
| *TPH2* rs4570625 | 5′-CCTCCATATAACTCTCATGAGGC-3′<br>5′-TCTTATCCCTCCCATCAGCATATT-3′ | [FAM]CACACATTTGCATGCACAAAATTAGAATATG [BHQ1]<br>[HEX]CACACATTTGCATGCACAAAATTATAATATG [BHQ1] |

The real-time PCR was performed in a 25 μL reaction mixture containing 15 ng of DNA, 100 nM forward and reverse flanking primers, 50 nM each of the two probes, and the BioMaster HS-qPCR Hi-ROX master mix (Biolabmix, Russia). The reaction began with denaturation at 95 °C for 600 s, followed by 36 cycles of 95 °C for 15 s, 54 °C for 30 s, and 72 °C for 30 s. Next, in the two plates analyzed first for each polymorphic site, 10% of the DNA samples were verified by Sanger sequencing. The obtained data were processed with LightCycler 96 software.

Statistical analysis consisted of an intergroup comparison of allele frequencies for each of the studied polymorphisms by Fisher's exact two-tailed test using SPSS 11.0 software (IBM Corp., Armonk, NY, USA). We compared groups 1, 2, and 3 and the pooled population "1 + 3". Differences were considered statistically significant at $p < 0.05$. In the case of multiple alleles, the following comparisons were made for the intron 2 VNTR of *SLC6A4*

(12 repeats vs. 9 repeats + 10 repeats) and for the exon 3 VNTR of *DRD4* (7 + 8 repeats vs. all others). To assess statistical significance in multiple comparisons, the significance correction of the Benjamini–Hochberg method was applied.

## 3. Results

Data on the number of genotypes for rs4680, rs4570625, rs258747, rs6265, and rs774676466 are given in Table 3, on VNTR STin2 of the *SLC6A4* gene in Table 4, and on exon 3 VNTR of the *DRD4* gene in Table 5.

All studied groups of subjects were in Hardy–Weinberg equilibrium (data not shown). There was no difference between the sexes in allele frequencies, and all the tested polymorphic sites are located on autosomes; therefore, males and females were combined for subsequent analyses. For polymorphisms STin2 VNTR and rs774676466 of the *SLC6A4* gene and rs4570625 of the *TPH2* gene (related to the serotonergic system), no significant differences in frequencies were found among the tested groups (Table 6).

For rs6265 (*BDNF* gene), differences were found between groups 1 and 2 ($p = 0.033$, $\chi^2 = 4.638$), with lower frequency of the presumed protective allele in group 2. Nonetheless, when the correction for multiple testing was applied, the statistical significance of the result disappeared (Table 6). The pooled "noncrisis" population (groups 1 + 3) did not differ from group 2 in this respect. A decreased frequency of the "risk" allele G of rs258747 (*NR3C1* gene) was noted in group 2 versus groups 1 and 3, but the difference was not significant (Table 6). Frequency of the "long" (7R + 8R) alleles of the *DRD4* gene was higher in the group of adolescents born in 1992–1995 (Tables 5 and 6). Groups 1 and 2 differed in this parameter ($p = 0.034$, $\chi^2 = 4.547$), but the significance was lost after the correction for multiple testing. Nevertheless, when group 2 was compared with the combined population of adolescents born before and after the crisis (groups 1 + 3), the differences were significant ($p = 0.020$, $\chi^2 = 5.492$; Table 6). A similar pattern was observed for rs4680 (*COMT* gene, which is also related to the dopaminergic system). The frequency of allele G was higher in "crisis" group 2 compared with group 3 ($p = 0.040$, $\chi^2 = 4.393$) and the combined "noncrisis" population (groups 1 + 3) (Table 6). The latter difference was significant ($p = 0.022$, $\chi^2 = 5.289$).

**Table 3.** Numbers of genotype carriers for rs4680, rs4570625, rs258747, rs6265, and the rs774676466 indel in the three groups of adolescents (1: born in 1982–1985, 2: in 1992–1995, and 3: in 2002–2005); n: the total number of carriers of all genotypes of a polymorphism.

| Group | rs4680 *COMT* | | | | rs4570625 *TPH2* | | | | rs258747 *NR3C1* | | | | rs774676466 *SLC6A4* | | | | rs6265 *BDNF* | | | |
|---|---|---|---|---|---|---|---|---|---|---|---|---|---|---|---|---|---|---|---|---|
| | AA | AG | GG | n | GG | GT | TT | n | AA | AG | GG | n | 16/16 | 16/14 | 14/14 | n | GG | AG | AA | n |
| 1 | 109 | 174 | 76 | 359 | 169 | 91 | 17 | 277 | 110 | 216 | 114 | 440 | 95 | 144 | 43 | 282 | 340 | 91 | 11 | 442 |
| 2 | 174 | 325 | 165 | 664 | 226 | 129 | 18 | 373 | 188 | 344 | 150 | 682 | 132 | 188 | 50 | 370 | 484 | 180 | 20 | 684 |
| 3 | 134 | 246 | 86 | 466 | 164 | 104 | 14 | 282 | 119 | 233 | 121 | 473 | 106 | 142 | 378 | 285 | 331 | 124 | 13 | 458 |

**Table 4.** Numbers of genotype carriers and allele frequencies for VNTR STin2 of the *SLC6A4* gene in the three groups of adolescents (1: born in 1982–1985, 2: in 1992–1995, and 3: in 2002–2005); n: the number of all such carriers in a group, R: the number of repeats.

| Group (n) | Number of Genotype Carriers | | | | | Allele Frequency | | |
|---|---|---|---|---|---|---|---|---|
| | 12R/12R | 12R/10R | 12R/9R | 10R/10R | 10R/9R | 12R | 10R | 9R |
| 1 (381) | 145 | 174 | 9 | 46 | 7 | 0.621 | 0.358 | 0.021 |
| 2 (654) | 244 | 332 | 15 | 54 | 9 | 0.637 | 0.343 | 0.018 |
| 3 (472) | 176 | 210 | 13 | 66 | 7 | 0.609 | 0.370 | 0.021 |

**Table 5.** Numbers of genotype carriers and allele frequencies for the exon 3 VNTR in the *DRD4* gene in the three groups of adolescents (1: born in 1982–1985, 2: in 1992–1995, and 3: 2002–2005); n: the number of all such carriers in a group, R: the number of repeats.

| *DRD4* | Group 1 n = 433 | Group 2 n = 666 | Group 3 n = 479 |
|---|---|---|---|
| No. of genotype carriers | | | |
| 2R/2R | 9 | 2 | 10 |
| 2R/3R | 2 | 3 | 1 |
| 2R/4R | 41 | 82 | 48 |
| 2R/6R | - | 1 | - |
| 2R/7R | 9 | 14 | 8 |
| 2R/8R | - | 1 | - |
| 3R/3R | 1 | 2 | 1 |
| 3R/4R | 22 | 40 | 17 |
| 3R/5R | - | 1 | - |
| 3R/6R | - | 1 | - |
| 3R/7R | 5 | 3 | 3 |
| 3R/8R | 1 | - | - |
| 4R/4R | 255 | 345 | 289 |
| 4R/5R | 14 | 16 | 10 |
| 4R/6R | 3 | 5 | 4 |
| 4R/7R | 56 | 125 | 76 |
| 4R/8R | 4 | 9 | 1 |
| 5R/5R | 2 | 1 | - |
| 5R/7R | - | 1 | 1 |
| 6R/6R | - | - | 1 |
| 6R/7R | 2 | - | 1 |
| 7R/7R | 6 | 11 | 8 |
| 7R/8R | 1 | 2 | - |
| 8R/8R | - | 1 | - |
| Allele frequency | | | |
| 2R | 0.080 | 0.079 | 0.080 |
| 3R | 0.037 | 0.039 | 0.024 |
| 4R | 0.751 | 0.726 | 0.766 |
| 5R | 0.021 | 0.015 | 0.011 |
| 6R | 0.006 | 0.005 | 0.007 |
| 7R | 0.098 | 0.125 | 0.110 |
| 8R | 0.007 | 0.011 | 0.001 |

**Table 6.** Minor allele frequencies of SNPs rs4680, rs4570625, and rs6265; VNTRs in exon 3 of *DRD4*; and VNTRs in intron 2 and indel rs774676466 in *SLC6A4* in the studied groups of adolescents (1: born in 1982–1985, 2: in 1992–1995, and 3: in 2002–2005) and *p* values; a "protective" allele was selected according to the literature data.

| Polymorphic Site | "Protective" Allele | Minor Allele | Minor Allele Frequency | | | *p* Value | | | |
|---|---|---|---|---|---|---|---|---|---|
| | | | 1 | 2 | 3 | 1↔2 | 2↔3 | 1↔3 | 2↔1 + 3 |
| exon 3 VNTR in *DRD4* | Long 7R + 8R | Long 7R + 8R | 0.105 | 0.136 | 0.111 | 0.034 | 0.074 | 0.707 | 0.020 * |
| rs4680 of *COMT* | G | G | 0.454 | 0.492 | 0.448 | 0.095 | 0.040 | 0.842 | 0.022 * |
| intron 2 VNTR in *SLC6A4* | Long 12 | Short (9 + 10) | 0.379 | 0.361 | 0.391 | 0.422 | 0.158 | 0.653 | 0.184 |
| rs774676466 of *SLC6A4* | Long 16 | Short (14) | 0.408 | 0.389 | 0.379 | 0.530 | 0.731 | 0.331 | 0.885 |
| rs4570625 of *TPH2* | T | T | 0.226 | 0.221 | 0.234 | 0.893 | 0.594 | 0.776 | 0.693 |
| rs6265 of *BDNF* | G | A | 0.128 | 0.161 | 0.160 | 0.033 | 1 | 0.053 | 0.212 |
| rs258747 of *NR3C1* | A | G | 0.504 | 0.472 | 0.502 | 0.141 | 0.163 | 0.925 | 0.086 |

* The result is statistically significant.

## 4. Discussion

Obviously, the response to prolonged stress is determined not by the carriage of certain alleles of individual genes separately but by the entire genome and epigenome; for example, among the polymorphic sites we analyzed, effects of interaction on phenotype formation have been previously documented for polymorphisms of *BDNF* and *COMT* [84], of *BDNF* and *SLC6A4* [66], and of *BDNF* and *TPH2* [85]. For rs774676466 (*SLC6A4*), rs6265 (*BDNF*), rs4680 (*COMT*), and the exon 3 VNTR of the *DRD4* gene, there is evidence of their co-influence with nearby genetic variants on the expression of encoded mRNA or a disease correlation [55,86–88]. Significant sex differences in the association of a genotype with predisposition to disorders have been found for the *COMT* gene [59]. In general, there are differences both in the regulation of the HPA axis and in the prevalence of stress-induced diseases between men and women [89–91]. In humans, for all the above genes (*SLC6A4*, *NR3C1*, *DRD4*, *TPH2*, *COMT*, and *BDNF*), promoter methylation is known to influence detectable stress markers or disease symptoms [87,92–96]. In the current study, we were unable to take into account genotypes and phenotypes of parents of the adolescents and could not determine the paternal or maternal origin of each allele. Nevertheless, our data made it possible to identify the imprint left by long-term stress in the gene pool of the generation born during this period and to identify genes that potentially may affect adaptation to long-term social stress. According to our data, the reproductive behavior of people during a socioeconomic deprivation period may be affected by the dopaminergic system. Similar conclusions about its significance have been previously made in individuals with low resilience to psychological stress [22,97]. It is known that the same alleles of common polymorphisms may be either risky or protective toward various disorders. This statement is also true for the VNTR in exon 3 of the *DRD4* gene and rs4680 of the *COMT* studied here. Long variants of the *DRD4* gene are more common among patients with ADHD [48]. In a meta-analysis of 363 datasets, ADHD and panic disorder proved to be associated with the G (Val) allele of rs4680 (*COMT*) among whites [62]. On the other hand, there is also evidence that rs4680 is not associated with ADHD [98]. Individuals with a smaller number of A alleles of rs4680 seem to report greater well-being and fewer depressive symptoms [99]. Homozygotes for the G/G genotype of this SNP are predisposed to internet gaming disorder and carriers of the G allele to drug addictions. This association is mediated by impulsivity and fun-seeking [100,101]. Of note, it is genes *COMT* and *DRD4* that are considered modulators of novelty processing [102].

Thus, frequencies of long alleles of the VNTR in exon 3 of the *DRD4* gene and of the G allele of rs4680 (*COMT*)—both associated with ADHD—are significantly higher in the generation born during the crisis of the 1990s in Russia than in the "noncrisis" generations. Nonetheless, the impact of the joint influence of epigenetic modifications and the genotype under social stress is still poorly understood. It is not known whether epigenetic modifications caused by prenatal or early-life stress are less or more common among carriers of "resilient" genotypes. Of note, homozygous carriers of the 7R allele of *DRD4* show more externalizing behavior and higher ADHD symptom severity after exposure to higher levels of prenatal maternal stress, whereas homozygotes for the 4R allele are insensitive to the effects of prenatal stress [103,104].

Possibly due to better adaptation to long-term stress (higher well-being and lower risk of anxiety-associated disorders), carriers of some variants of the dopaminergic system genes may show subdominant reproductive behavior less often and have more offspring on average during periods of severe socioeconomic deprivation.

## 5. Conclusions

The observed difference between generations in two variants of the dopaminergic system genes may be due to differences in reproductive behavior among carriers of distinct genotypes during periods of prolonged socioeconomic crises. These data require replication in other populations.

**Author Contributions:** Conceptualization, S.V.M.; methodology, D.E.I. and M.S.A.; validation, L.V.S. and P.S.O.; formal analysis, D.V.D.; investigation, D.E.I., E.A.Y. and A.B.; data curation, S.V.M.; writing—original draft preparation, S.V.M.; writing—review and editing, S.V.M.; project administration, S.V.M. All authors have read and agreed to the published version of the manuscript.

**Funding:** This research was funded by Russian Science Foundation grant # 22-28-00866 "Study of the genetic dynamics of the urban population during the socio-economic crisis".

**Institutional Review Board Statement:** The study was conducted in accordance with the Declaration of Helsinki and approved by the ethical committee No. 56 of the Institute of Internal and Preventive Medicine, a branch of the Institute of Cytology and Genetics of the Siberian Branch of the Russian Academy of Sciences, Novosibirsk, Russia (Protocol No. 7, approved on 22 June 2008).

**Informed Consent Statement:** Informed consent was obtained from parents and legal guardians of all the subjects involved in the study. Written informed consent for publication of the study results was obtained from them as well.

**Data Availability Statement:** The data presented in this study are available upon request from the corresponding author.

**Acknowledgments:** The authors thank L.I. Kuchirka for technical support.

**Conflicts of Interest:** The authors declare no conflict of interest. The funders had no role in the design of the study; in the collection, analyses, or interpretation of the data; in the writing of the manuscript; or in the decision to publish the results.

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
