# Peer review of "Prevalence of Common Alleles of Some Stress Resilience Genes among Adolescents Born in Different Periods Relative to the Socioeconomic Crisis of the 1990s in Russia"

_cimb, doi:10.3390/cimb45010004_

Round 1

Reviewer 1 Report

Authors analyzed polymorphisms of 6 genes with PCR and subsequent Sanger sequencing. Based on specific differences in frequencies emanated from the resulted analyses, they extrapolated that these differences in the identified genes may have had impact on reproductive behavior or successful adaptation of a person to social stress.

Although the overall quality is above average with English expression and grammar in check, the methods used do not meet the criteria to present a molecular and genetic epidemiolocal study. Sanger sequencing has been replaced by next-generation sequencing (NGS) and currently third-generation sequencing (long-read sequencing) is under development.

Presumptions made cannot be supported by the methodology used.

Author Response

 Dear Reviewer, we appreciate your interest in our study.

1. Although the overall quality is above average with English expression and grammar in check, the methods used do not meet the criteria to present a molecular and genetic epidemiolocal study. Sanger sequencing has been replaced by next-generation sequencing (NGS) and currently third-generation sequencing (long-read sequencing) is under development.

Presumptions made cannot be supported by the methodology used.

Most of the studies in population molecular genetics, in which the frequencies of already known gene variants are analyzed, are carried out using PCR methods. In this work, Sanger sequencing was not used for genotyping, but to confirm the correct interpretation of the obtained PCR data. This paper is part of a larger work in which half of the studied polymorphic loci will be VNTRs. (3 out of 7 polymorphic loci in this manuscript). It is known that the analysis of tandem repeats by sequencing is complicated. For example,

  • Bakhtiari M, Shleizer-Burko S, Gymrek M, Bansal V, Bafna V. Targeted genotyping of variable number tandem repeats with adVNTR. Genome Res. 2018 Nov;28(11):1709-1719. doi: 10.1101/gr.235119.118. Epub 2018 Oct 23. PMID: 30352806; PMCID: PMC6211647.

Genotyping VNTRs in a donor genome sequenced using short (Illumina) or longer single-molecule reads, requires the following: (1) recruitment of reads containing the VNTR sequence; (2) counting RUs for each of the two haplotypes; (3) identification of indels within VNTRs; and (4) identification of mutations within the VNTR. Mapping tools such as BWA (Li and Durbin 2009) and Bowtie 2 (Langmead and Salzberg 2012) can work for read recruitment for STRs, but are challenged by insertion/deletion of larger repeat units. Mapping issues also confound existing variant callers, including realignment tools such as GATK IndelRealigner (DePristo et al. 2011) if the total VNTR length is larger than the read length. This is because reads contained within the VNTR sequence have multiple equally likely mappings and therefore will be mapped randomly to different locations with low mapping quality (Kirby et al. 2013). Detection of point mutations in long VNTRs requires integrating information across the entire VNTR sequence. For VNTRs whose total sequence length (RU count times the RU length) is much longer than the read length, detection of SNVs and indels is not feasible using existing variant callers.

  • Bakhtiari M, Park J, Ding YC, Shleizer-Burko S, Neuhausen SL, Halldórsson BV, Stefánsson K, Gymrek M, Bafna V. Variable number tandem repeats mediate the expression of proximal genes. Nat Commun. 2021 Apr 6;12(1):2075. doi: 10.1038/s41467-021-22206-z. PMID: 33824302; PMCID: PMC8024321.

Despite their importance, the full extent of VNTRs in mediating Mendelian and complex phenotypes is not known due to genotyping challenges. Traditionally, VNTR genotyping used capillary electrophoresis which did not scale to large cohorts. Despite the advent of sequence based genotyping, repetitive sequences continue to be challenging for genomic analysis. For example, “stutter errors” due to polymerase slippage during PCR amplification change VNTR length and reduce genotyping accuracy. While tools for genotyping STRs have been developed, they generally do not detect or genotype VNTRs, which have non-identical and larger repeat units. Recently, a few specialized computational methods (including our own method, adVNTR) have been published to tackle the problem of genotyping VNTRs from sequence data. However, these methods are too computationally intensive to scale to functional studies with hundreds of individuals and 104 VNTR loci (Results). There have also been recent, successful efforts to genotype VNTRs using long-read sequencing technologies such as Pacific Biosciences (PacBio) and Oxford Nanopore Technologies (ONT). While these methods (which include adVNTR) are quite accurate, the technologies are currently too expensive for population scale sequencing.

Therefore, in order to save time on interpreting the results, we used the PCR, RT-PCR, and PCR-RFLP methods in our work.

Reviewer 2 Report

The paper delivered by Mikhailova et al. analyzes the genotypes of adolescents born before, during, and after the 1990 crisis in the city of Novosibirsk, Russia, to assess the genetic differences between these generations and their association with the genes responsible for reproductive behavior. The study is well conceived and well executed. This reviewer is pleased with the importance of this study, the care with which it was conducted, and the implications of the results for human health. However, while the results presented are convincing, the work raises some minor concerns that will need to be addressed:

1)    The introduction is extensive, so it must be summarized, mainly the specific function of each gene. In addition, the paragraph on line 67 and the one on line 182 seem to repeat the purpose of the work. These ideas could be brought together to improve the writing.

2)    Define the abbreviation CNS.

3)    The numbering of all the tables is incorrect, both in the table and text. Table number 1 should be the one with the name: “Conditions for PCR and restriction analysis for the VNTR of exon 3 in DRD4, VNTR STin2 and rs774676466 in SLC6A4, and rs4680 of COMT. R: the number of repeats” and it would have to be quoted on line 217

4)    Although the last paragraphs could work as a conclusion of the work, it is not clear. Therefore, the conclusion must be delimited. It is suggested to write it clearly in another section or use the phrases: "in conclusion" or "in summary."

Author Response

Dear Reviewer, thank you very much for the thorough review of our paper.

1)    The introduction is extensive, so it must be summarized, mainly the specific function of each gene. In addition, the paragraph on line 67 and the one on line 182 seem to repeat the purpose of the work. These ideas could be brought together to improve the writing.

We added summary information (lines 180-185)

We removed the paragraph on line 67

2)    Define the abbreviation CNS.

We have changed the abbreviation to "central nervous system" because it is only used twice in the text.

3)    The numbering of all the tables is incorrect, in both the table and text. Table number 1 should be the one with the name: “Conditions for PCR and restriction analysis for the VNTR of exon 3 in DRD4, VNTR STin2 and rs774676466 in SLC6A4, and rs4680 of COMT. R: the number of repeats” and it would have to be quoted on line 217

We inserted the correct table numbering (lines 215, 219, 226, 228, 246, 247, 249, 253, 257, 266, 267, 275, 278, 279, 283, and 286)

4)    Although the last paragraphs could work as a conclusion of the work, it is not clear. Therefore, the conclusion must be delimited. It is suggested to write it clearly in another section or use the phrases: "in conclusion" or "in summary."

We have added a Conclusion section (lines 335-339)

Reviewer 3 Report

The paper is worth publishing in the present form. Some more comments about the possible links between stress and other diagnoses, beyond adhd, are welcome if supported by any background data.

Author Response

Dear Reviewer, we appreciate your time and your interest in our study.

Some more comments about the possible links between stress and other diagnoses, beyond adhd, are welcome if supported by any background data.

There is a great deal of evidence linking stress with metabolic, cardiovascular, and immunological disorders. It is difficult to present this information briefly in this paper. With regard to psychiatric disorders, ADHD is of particular interest because it has a high heritability (75%), i.e. less than other violations depends on environmental factors. We focused on ADHD since this pathology is common for the DRD4 and COMT genes, for which differences in allele frequencies were found.

Round 2

Reviewer 1 Report

There was no hint at using Sanger sequencing as a tool for genotyping. Authors' response is evasive, lacking the essential providence to address the major weakness of the current submission.

Author Response

Dear Reviewer, thank you for your attention to our paper.

In our work, Sanger sequencing was not used for genotyping, but only for checking the accuracy of real time PCR results, we added a clarification (line 234-235). It is Sanger sequencing that is used to validate NGS results because it is more accurate, although less productive. In our study, the use of NGS would be redundant for population genetic analysis (more expensive, less accurate for VNTR, and more laborious) than combining real time PCR with Sanger sequencing validation at the outset. This study design is widely accepted and described.

Round 3

Reviewer 1 Report

Nothing to disclose.